# Confronting Racism within the Canadian Healthcare System: Systemic Exclusion of First Nations from Quality and Consistent Care

**DOI:** 10.3390/ijerph17228343

**Published:** 2020-11-11

**Authors:** Wanda Phillips-Beck, Rachel Eni, Josée G. Lavoie, Kathi Avery Kinew, Grace Kyoon Achan, Alan Katz

**Affiliations:** 1Department of Community Health Sciences, College of Medicine, First Nation Health and Social Secretariat Manitoba and the University of Manitoba, Winnipeg, MB R3B 2B3, Canada; wphillips-beck@fnhssm.com; 2Independent Researcher, Victoria, BC V9C 0M1, Canada; 3Department Community Health Sciences, College of Medicine, University of Manitoba, Winnipeg, MB R3E 3P5, Canada; Josee.Lavoie@umanitoba.ca; 4First Nation Health and Social Secretariat Manitoba, Winnipeg, MB R3B 2B3, Canada; kathikinew@gmail.com; 5Education Indigenous Institute of Health and Healing, University of Manitoba, Winnipeg, MB R3E 3P4, Canada; Grace.KyoonAchan@umanitoba.ca; 6Manitoba Centre for Health Policy, Department Community Health Sciences and Family Medicine, College of Medicine, University of Manitoba, Winnipeg, MB R3B 2B3, Canada; Alan_Katz@cpe.umanitoba.ca

**Keywords:** racism, Canadian healthcare system, First Nations, grounded theory

## Abstract

The study is on racism against First Nation peoples in the Canadian healthcare system. The study design incorporates principles of grounded theory, participant and Indigenous (decolonizing) research. Four questions are addressed: (1) What is the root cause of racism against First Nation peoples in the healthcare system? (2) What factors perpetuate racisms existence? (3) What are the impacts of racism on First Nation health? (4) What needs to be done to eradicate racism and to create an equitable healthcare system that sufficiently represents the needs, interests and values of First Nation peoples?

## 1. Introduction

The focus of this paper is on racism against First Nation peoples in the Canadian healthcare system. The discussion is based on a grounded theory analysis of interviews and focus group discussions with participants of a research project on community-based primary healthcare supporting transformation in the health of Manitoba First Nations. Analysis takes place within a macro-context of a global pandemic, and international protests against institutional racism.

The study design is heavily based in partnership development and principles of participatory research. The goal of engagement and discovery is implementation in order to transform healthcare into a system that is equitable, accessible, and appropriate to First Nation peoples. The study addresses four questions: (1) what is the root cause of racism against First Nation peoples in the healthcare system? (2) What factors perpetuate racism’s existence? (3) What are the impacts of racism on First Nation health? (4) What needs to be done to eradicate racism and to create an equitable healthcare system that sufficiently represents the needs, interests and values of First Nation peoples?

### 1.1. Contextualizing Racism within Canadian Healthcare

This year, racism in the USA against Black Americans took centre stage when a White police officer brutally and publicly murdered an unarmed Black man in the streets of Minneapolis whilst other officers stood there and watched. A worldwide eruption of outrage and support for racial equality and justice ensued. It didn’t take long for the issue of police brutality against Black and Indigenous peoples in Canada to garner its long-needed attention. At the top, Canadian Prime Minister Justin Trudeau admitted in a public address that, “as a country, we can’t pretend that racism doesn’t exist here… unconscious bias is real and systemic discrimination is real and they happen here in Canada” [1].

The worldwide movement against systemic racism is taking place within the backdrop of the COVID-19 global pandemic. Speaking at the 18th Nelson Mandela Annual Lecture in Johannesburg on 18 July 2020, United Nations Chief Antonio Guterres stated that the pandemic is likened to an X-ray, revealing fractures in the skeleton of the societies we have built.

It is exposing fallacies and falsehoods everywhere. The lie that free markets can deliver healthcare for all, the fiction that unpaid care work is not work, the delusion that we live in a post-racist world, the myth that we are all in the same boat. Because while we are all floating on the same sea, it is clear that some are in super yachts, while others are clinging to drifting debris [2].

Evidence reveals that racism exists within the Canadian healthcare system [3,4]. Though touted by Canadians to be one of the best in the world, the Canadian Public Health Association acknowledges Canada as a nation where race, culture and religion are persistent determinants of health inequities. Effects are compounded for Indigenous peoples [5].

The health gap between First Nations and other Canadians narrowed in the 1970s and then widened again. A recent report by researchers at the Manitoba Centre for Health Policy found that the gap is widening still [6]. Racial injustice is constructed through societal notions of white supremacy, racism, ethnocentrism and discrimination and their effects on First Nation health. Through stories gathered for this study, we examined interpersonal, systemic, environmental, and unconscious or implicit bias, and their intersections. We implemented an expanded gaze from healthcare to the Canadian governance structure. We sought further to the dominant worldviews and ideological systems, as disseminated through official political-economic discourses and impinging upon day-to-day lives in communities. Finally, we expounded upon imperatives of a colonial mindset to greater inclusion and at least equal consideration of First Nation worldviews—from physical to relational worlds, in order for our research to move beyond the stuck points, beyond studying discrepancies, gaps, in-accessibilities and other-ness, towards inclusion, social justice, healthy diversity, and equity.

### 1.2. Examining the Multilevel Forms of Racism

The Canadian Prime Minister used the terms “unconscious bias” and “systemic racism” in his address, noted in the introduction of this paper [7]. The former is recognized in the social sciences as a central tenet of qualitative inquiry. It regards any influence on behalf of a researcher’s perspectives that result in a distortion to the topic under study [8]. Unconscious bias, also known as ‘implicit bias’ or ‘implicit social cognition’ is a system of beliefs or stereotypes that influence our attitudes, behaviours, and decisions, unconsciously [9,10]. Unconscious bias emanates from our upbringing and from images repeatedly projected upon us via the media, governments, educational institutions, and our social circles. Science on the study of unconscious bias is extremely disturbing in terms of how our minds categorize and exclude people outside of our own racial groups [11]. Studying various interactive situations in the USA, a leading psychologist on the topic confessed she was shaken by her own findings that the concept isn’t simply a bias where one thinks, ‘this group is not as good as my group.’ She explains that it is like placing others, i.e., African Americans, outside the human family altogether [12]. Unconscious bias on the part of healthcare providers may manifest in differential diagnosis and an inability to see patients beyond the colour of their skin, ultimately to the detriment of patient health and life. Unconscious bias happens to individuals within a racist society [13].

Systemic racism can be traced back to contact between the First Nations and European colonizers. Martinez writes, systemic discrimination has impacted Indigenous health detrimentally, dating back to the late 15th century when French and British expeditions explored, colonized and fought over various land areas across, what is today, Canada. The colonial narrative of cultural hierarchy and white supremacy exists today, translated over centuries into gaping health disparities. White supremacy is an historically-based, institutionally perpetuated system of exploitation and oppression of continents, nations, and peoples of colour by white peoples and nations of the European continent, for purposes of maintaining and defending systems of wealth, power, and privilege [14]. Researchers on health, social and cultural impacts of hydroelectric developments in Manitoba and Quebec have long connected a desire for land and natural resources for the accumulation of wealth and power and the subjugation of First Nations necessitated within a capitalist and colonizing Canadian system [15,16].

Racism is often manifested with more subtlety than violent verbal or physical attacks. It can be implied through actions as well as attitudes and through a lack of consideration to such things as safety and comfort of others deemed undeserving. Racism can be revealed through heightened suspicion or distrust of a people based on perceived racial difference. DiAngelo, in her book on ‘White Fragility’, writes that in the Western colonial context, white people hold institutional power. This implies an understanding that racism is more than the sum of actions and slurs, it is a system—it is prejudice plus power, designed to benefit and privilege whiteness by every economic and social measure. Everyone has unconscious bias, but when one supports a people’s collective bias with lasting authority and institutional control, it is vastly transformed [17]. Systemic racism, also known as institutional racism, refers to established laws, customs, or practices that are systematically reflected in and that produce racial inequities in society. Whether it is overt or unintentional or stems from oppressive or negative race-based policy, systemic racism contributes to health and socio-economic disparities, a greater exposure to risk, hazards, toxic environments, unfair perceptions, treatment, and injustices all of which ultimately influence health [18]. Further, racism can be destructive in its ability to obscure, i.e., in his book, ‘A Review of Racism Without Racists: Color-Blind Racism and the Persistence of Racial Inequality in America’ (4th ed.), Bonilla-Silva points to the fact that in today’s social climate, most ‘Whites’ disassociate themselves with the idea of being racists. Rather, he states, that in the United States, they hide behind issues of race and race inequities with a declaration that they, “don’t see color, just people.” Bonilla-Silva’s book focuses on the problems reinforced by colour-blind racism, reflected in racialized social systems and which reinforce the status quo [19].

## 2. Theoretical and Empirical Context

### 2.1. Establishment of Colonial Control

History paves the road to expectation and unless transformed, expectation is unchanged from what a people experienced in the past. To conceive of racism within healthcare and its impacts on the health of First Nations requires an understanding of their historical relationship with the governments of Canada [20,21]. Nearly 150 years ago, the Canadian government created the Indian Act 1876 [22] with an aim to destroy tribal governance systems and absorb the First Nations into the Canadian mainstream. It imposed restrictions upon them as a means to subjugate a race. It took the power to establish identities and to confer rights and status as befits the Crown. Individuals, families, and communities were divided according to those who were to be considered enfranchised under the law, those who would have tribal (called ‘band’ in the Canadian context) affiliation, and those who would not. The Indian Act is, by nature, a racist document, designed to eradicate First Nation peoples out of existence [23,24].

Under the Indian Act, women were denied status, residential schools and reserves were established, and individuals were renamed. People were confined to small plots of communal land assigned to them and called reserves, access to traditional foods were declined, and they were prohibited from forming political organizations or practicing traditional spiritual and healing ceremonies. First Nation peoples who attempted to practice their healing techniques were imprisoned, and up until the 1950s, had their medicine bundles confiscated, put on display in museums or private collections, or destroyed [25,26]. Each of these exclusions and denials were purposefully designed to assimilate or “civilize” First Nation peoples and to destroy cultures, ways of being, out of existence.

Alternately, the Indian Act serves as a legislative tool that holds the federal government accountable for legal responsibilities to the First Nations. There are economic, education and health provisions within the Act. Further, it serves to protect the First Nations, at least somewhat, from interference by the provinces. Thus it should not be abolished without commitment to a tangible plan for self-government that acknowledges First Nation rights [27,28,29]. Essentially, the Indian Act provides an alternative governance infrastructure to the one that was destroyed, and this governance structure has now been entrenched, making its elimination complex.

### 2.2. First Nation Healthcare

Jurisdictional gaps continue to affect a seamless delivery of healthcare services to the First Nations [30] and although the provinces and territories deliver a complement of services spanning primary healthcare to hospital care through a publicly funded universal system, various social and geographic factors exclude accessibility to First Nations [31,32]. The decentralized nature of the Canadian system, which is managed by each of its ten provinces and three territories, results in considerable variations in the configuration of services provided. For example, allied health services such as physical or other therapies, dental care for those with low income and children, and subsidies for expensive medications may or may not be included in provincial/territorial services. Coverage for these services may require private health insurance or be paid out of pocket. The federal government professes that it supports First Nations to reach a reasonable level of access to care that is comparable with the rest of Canadians, through Indigenous Services Canada’s First Nations Inuit Health Branch (FNIHB). FNIHB funds Indigenous-centric programs (the Non-Insured Health Benefits Program, Healthy Child, Mental Health and Addictions, and Environmental Programs [33], providing for a limited range of supplementary services and resources not available elsewhere. In fact, FNIHB has repeatedly asserted their intention to assume the role of ‘payer of last resort’ [34].

From the perspective of the First Nations, none of the historical treaties stated that there was to be an exchange between land and resources—they all stress that Treaties were to share the land, but that it was First Nations land. Healthcare was to be provided by the government of Canada as protection from settler infection and diseases, for example, as mentioned in the medicine chest clause in Treaty 6. The First Nations were to keep their own governance and medicines, and as the Treaties were silent on replacing those, so it remains. It was the Indian Act, consolidated under then Prime Minister John A. MacDonald (in the mid to late-1800s), and amended by subsequent Prime Ministers and governments that deliberately misinterpreted the Treaties and used this Indian Act to contravene and totally undermine First Nation sovereignty.

The Canadian public is not properly informed about the nature of the relationship and exchange between the federal government and First Nations. Notions that the First Nations receive all sorts of ‘freebies’ when it comes to education and health resources exist. These misconceptions have deleterious impacts to health [35].

### 2.3. Racism in Healthcare

Recent reports draw attention to racism’s prevalence within the healthcare system. Examples, like the stories of Brian Sinclair who, in 2008, died in the waiting room of Winnipeg’s Health Sciences Centre after waiting 34 h to be seen for a treatable illness, which would have required a fairly minute procedure [36], and the many women in Saskatchewan who received tubal ligations without their consent, are examples of blatant racism against Indigenous peoples and are numerous [37]. Understanding effects of racism on health behaviours requires an understanding of the history of intergenerational trauma caused by the residential school system with what Boyer called “it’s myriad tentacles of physical and sexual abuse” [38] and other policies rooted in colonialism. History has established a platform upon which a substantial power imbalance between healthcare providers and First Nation patients may persist [39,40,41,42,43,44,45].

To fully appreciate the extent to which racism persists and perpetuates itself throughout healthcare requires an analysis of the structures of the discourses upon which healthcare is implemented. Political cognitions written as official discourses are essentially the products of complex inter-elite influences of politicians echoing and furthering ideas put forth by scholars and other experts, who are members of the colonial elite. Policies constructed from these discourses are regurgitated from multiple, dependent influences of the limited few that have access and are reflected in the authoritative framework of Canadian governance [46]. Within the discourses, some voices are silenced and some issues are selected for discussion while others are ignored. Further, those who are *othered* [47] by the discourses tend, themselves, to respond according to script. For example, someone who is First Nation may, as a result of these maneuverings of colonization, doubt themselves in an interaction with a physician (perceived expert) in a conversation about their health. Deconstructing the imbalances of power and systematically exclusionary characteristics of our political discourses may shed light upon spaces that hold the roots of racism and where racism may be disseminated.

Further, discourses contain values, which are fundamental beliefs guiding or motivating attitudes and behaviours, and aiding in determination of what is important, good, desirable, or worthwhile in the world. Ethical differences are crucial components in understanding what might be behind what we call ‘unconscious biases’ of the dominant society.

## 3. Methodology

The current study is one of several research focuses of the Innovation in Community-based Primary Healthcare Supporting Transformation in the Health of First Nation and Rural/Remote Communities in Manitoba (iPHIT) project. The 5-year project, led by the First Nations Health and Social Secretariat of Manitoba (FNHSSM) and 8 Manitoba First Nation communities in collaboration with researchers at the University of Manitoba, aimed to work with communities that have developed different primary healthcare models and identify key elements for success from the perspectives of the First Nations regarding program delivery. Results were collated and workshopped with the participating Manitoba First Nations to support the development and implementation of the models in other communities across the province in order to bring about better health outcomes. A more complete description of the community-based, participatory approach of the overall iPHIT project is available elsewhere [48].

iPHIT committed to identifying and responding to ethical challenges that might have arisen through community-based participatory research methods, particularly with respect to different interests, priorities and values of the university and the communities. FNHHSM provided advice on First Nation research protocols and coordinated an ethics submission to the Health Information Research Governance Committee (HIRGC), a First Nations regional committee that is supported by FHHSSM at its secretariat and is responsible for ethical review of all proposals involving First Nations in Manitoba. HIRGC and the University of Manitoba Health Research Ethics Board (UM-HREB) provided ethical approvals for this and all studies defined under iPHIT. Ethical engagement within the project entailed an ongoing and fluid commitment where the responsibility for awareness, understanding and adherence to community ethics and protocol requirements rested with the researchers.

Community engagement is a process of inclusive participation supporting mutual respect of values, strategies, and actions for authentic partnership of people affiliated by special interest, cultural background, geographic proximity or similar interests to address issues of wellbeing [49]. Community engagement was implemented in terms of three distinct conceptual levels with a goal of ensuring that individuals and communities who were the focus of the study were also those guiding the study’s methodological process. Specifically, all First Nation partners were integrated in all of the iPHIT studies, fulfilling multiple roles as team members, research leads, research assistants, advisors, data collectors and knowledge users, providing key stewardship of the study process and outcomes. A First Nation nurse program and practice advisor for the FNHSSM coordinated the study. The partnering First Nations each hired a local research assistant to work on the overall project for the first 3 years of iPHIT development and implementation. Finally, the research was unfolded in appreciation of the imperative to open up the spaces within which traditional academic research happens, that is to ‘de-colonize’ the physical and theoretical spaces within which research is performed. In the context of our research, to decolonize means to challenge Eurocentric research methods that have undermined Indigenous knowledge and experience by placing their voices and epistemologies in the centre of the research process [50,51]. The team had hoped that in preparing for a de-colonizing approach to research, flexibility and collaboration in the process of co-creating knowledge would be assured. The research design intended for fluidity of the research spaces, so that researchers/participants could feel equally at ease at the university, FNHSSM offices and in the communities. In reality, each of the collaborators would feel different levels and degrees of comfort in spaces depending upon feelings that one ‘belongs’, is employed by, or has their interests and backgrounds reflected within the different locations.

Communities included in the project represent four of 5 Manitoba-region Indigenous languages, i.e., Cree, Dakota, Dene and Ojibway, and all geographic variations, i.e., isolated without road accessibility, semi-isolated, i.e., access to seasonal roads, and non-isolated. The province’s different community care models were represented, i.e., nursing stations, funded either by federal or provincial governments, and community health centres, which operate through FNIHB funding allocations and are managed by the First Nations. The communities varied in terms of “transfer agreements” with the federal government, via FNIHB, according to stages of increasing self-governance [52]. Communities are said to progress, owning greater ownership and flexibility of their healthcare services and program designs and delivery, a characterization by the healthcare system that is itself problematized within the current study.

### 3.1. Grounded Theory—Theory Induced from Empirical Data

The study was conducted according to the principles of Grounded Theory (GT) as described by Charmaz [53,54]. The approach focuses on social processes or actions and inquires about interactions between people, and the outcomes of those interactions. GT was implemented to attempt to reveal the influences of symbolic internationalism as a social psychological approach that focuses on creating meaning from human interactions. Following others [55], we began the study with open questions and conversations as we attempted to learn more regarding the meanings behind the thoughts and actions of the research participants about racism. Establishing trust and allowing for open and uncomfortable conversations to take place provided opportunities for us to point out unconscious bias or systemic racism within our work.

GT cannot be studied *a priori*, as it is an approach not emanating from, but rather, in discovery of theory [56]. GT is unlike conventional quantitative methodologies where the goal of the literature review, mainly, is to refine the research question, determine the state of knowledge of the topic, determine gaps in the existing research, and identify a suitable design and data collection method [57]. In a GT approach, when and how the literature search is conducted is more ambiguous and often reliant upon empirically derived data, first. In fact, it is often suggested that researchers conduct the literature review following data collection and multiple reviews of the data. The literature review for the current study was refined several times following repeated reviews and greater depth of analysis of the data [58].

In a GT study, concepts under examination are generated from the data and not from existing literature. Likened to a detective striving to explain what is actually happening, the GT researcher strives to explain the main concerns of participants on a specific topic/situation and to discover how they resolve and/or process the concerns. The aim is to avoid preconceptions, remain open to different worldviews, ideologies, coping mechanisms, etc. The process is not an easy one, particularly for academics who are trained to read a lot, to define areas of interest according to the rules of science and to build new knowledge from previous empirically published findings in their field [59].

### 3.2. Recruitment of Focus Group and Interview Participants

Various meetings and focused discussions between researchers (at the University and FNHSSM) and First Nations Chiefs, Councils, Elders, health directors, and other community members initiated iPHIT development and implementation. Local research assistants (LRAs) from each First Nation recruited interviewees via a snowball sampling approach. LRAs took the lead role in developing the interviews, setting up interview times and places (typically in interviewee homes or at the health centres). They obtained research consents and assisted in analyses and interpretation of the data.

## 4. Findings

Findings are based on 299 in-depth interviews and 8 focus group discussions with First Nation men and women living in 8 Manitoba First Nation communities held 2013–2015. Participants were community-based health directors and staff, Elders and other community members, and users of the healthcare system. Interviews and focus group discussions were audio-recorded and transcribed. Analyses were performed utilizing GT techniques for categorization and coding of qualitative data. Transcriptions were imported into NVIVO software, which assisted in the management and organization of data. Excel was also used to organize observations from the data and to record emergent themes. Transcripts were checked for accuracy against audio-recordings. A second phase of analysis included line-by-line coding and use of the constant comparative method, a process by which newly collected data is compared to former, existing data to derive new codes, themes, and conceptual focal points.

Ultimately, rich data was organized into themes. Greater depth of reading into the data led to a re-organization and reshaping of the themes, until ultimately, all of the co-researchers felt that the data was organized as a true reflection of the meaning of the experiences of racism in the healthcare system of the research participants. Themes inducted from the data are: (1) a history in disrepair, (2) navigating a system that devalues First Nation health and wellness, (3) a preference for English and biomedicine, (4) fundamental systemic racism, (5) the racist alcoholism narrative, (6) poor quality service attributed to racism, (7) seeking racial and cultural similarity, (8) few opportunities for meaningful engagement and familiarity between First Nation and non-indigenous Canadians (healthcare providers), and (9) lack of consideration of accessibility to healthcare issues for First Nations.

### 4.1. A History in Disrepair

There is the history of distrust by First Nations towards Canadian institutions, as well as of the general population. Ongoing interpersonal conflicts and unresolved trauma stemming from residential schools and an overall history of Canadian colonization of First Nation peoples seeped through some of the stories that were shared.

Because of the history we had with the outside influences here residential schools, health services, and so on, it’s been passed from generation to generation not to trust people and because of the diseases that the European settlers brought with them. And still, that sticks today, those types of things.(E011)

I found the community doesn’t use enough of the services in the Regional Health Authority [an agency of the provincial government] because of the mistrust of the greater community around us. The dominant community, that’s meaning the health services, in local hospitals and all those places, I find they are here and though they are free, we would [rather not use them].(E011)

Because of the history of our two people Mistrust and incidents, like violence in their hospitals Still, some of our community members go into the hospitals and there is mistrust and they too are scared of us.(E011)

The participant quoted below alluded to the incredible amount of work required in order to fix the history and to move forward in building health equity:

(We are all) trying to (re) build that trust because the trust was broken, once it’s broken it’s a very hard thing to repair.(F019)

### 4.2. Navigating a System That Devalues First Nation Health and Wellness

Participants provided examples from throughout the system where their needs were either unmet or were placed secondary to some other interest, e.g., financial or ongoing jurisdictional debate. The sum impression of the comments is that the healthcare system is one that, for a myriad of different excuses, fails to promote or treat First Nation health concerns.

There’s always the argument that there’s not enough money or we can’t say certain things because we don’t want to sound ungrateful or get our funders angry. Well, wait a minute! You (the federal government and related organizations) are there for advocating First Nation peoples. Your job is to advocate for us and I don’t care about funding.(D014)

Our own members are literally given a juggling act because they go and receive health treatment, they get a prescription and the pharmacist asks them if they’re Status Indian and they say ‘yeah’ and they provide their information. And it’s not covered by non-insured health benefits. So then, well, if they’re living in Winnipeg, they are on social assistance program with the province, ‘we’ll try through the province.’ It doesn’t get approved because the province says, well, wait a minute, you’re a Status Indian; you’re federal responsibility so back and forth and then the members end up having to pay themselves. It’s a system that needs to be fixed and it affects us all in the community. We are left hanging in the balance.(D014)

Treating First Nation health issues, without consideration to social-economic, cultural and geographical conditions often leaves patients in uncomfortable, or even dangerous circumstances.

When we go out to the medical appointments, [we are] sent out alone and our stay is not covered. We’re in a strange place, especially me, I don’t know my way around the city it’s scary out there when they do that to our people you know.(G002)

You don’t get an escort when you’re sent out (to the city for medical appointments), Culture shock you call that. Oh where am I? It’s difficult when you don’t have you don’t know the resources in the city you’d be having a culture shock too coming into my community.(GFG004-3)

When my wife leaves when she was pregnant the last two times it was hard to go away from this community for that long period of time and to be in the city with your wife, and you and the kids to try and live there for 2 or 3 months, sometimes even 4 for some people. And it’s hard to live in Winnipeg because a lot of those services there don’t accommodate you you’re from an isolated community it’s sometimes, I can’t really go with her because those people say she doesn’t need an escort.(C011)

### 4.3. A Preference for English and Biomedicine

Participants commented on the preference for English and biomedical wisdom and its effects of First Nation patients throughout the system.

They know what they’re doing but you don’t know because they’re not communicating anything with you.(D017)

The older people they don’t understand English they don’t know how to read or write. And they don’t understand what you’re saying to them. They’re speaking English and they don’t understand that.(G001)

The participant in the quote below blames physicians for preferring the information in their texts to information they can collect through engaging in conversations with the First Nations:

The medical profession can come and talk to us, because you cannot learn anything from us unless you ask. You ask in a good way, we will share, but you cannot believe anything in books, like, just technical.(GFG004-11)

In the next comment, the participant offers a solution to improve communication and understanding between First Nation patients and a non-Indigenous medical system:

Here in the community what’s needed most is an advocate… because somebody that doesn’t know English properly, or even medical terms, stuff like that, can come in here and just sit Maybe a person has a cut somewhere or some kind of infection so you need a person that can translate, talk to this person they trust and say, ‘Can you tell that nurse what is wrong with me?’.(C010)

### 4.4. Fundamental Systemic Racism

The participants quoted below allude to two characteristics of a racist society, specifically, destruction of traditional First Nation medical wisdom and the Canadian governments preference to responding to the interests of non-Indigenous peoples.

We have developed a complete dependency and people have forgotten when people start to wake up that’s when they are going to start to heal. People don’t understand that, the emptiness of their feeling is the loss of purpose, the loss of spirit: our traditions.(D012)

When you’re talking about healthcare, you’re talking about the wellbeing of a person, you’re talking about the human right and the Canadian right to proper healthcare that is a fundamental value we have in Canada our proudest achievement, health for all it’s not the same for First Nation peoples. It’s a lower standard a slower process because of all the bureaucracy that our people have to go through and it’s impacting our health in a major way. We have to educate the rest of Canada about our health issues.(D014)

### 4.5. The Racist Alcoholism Narrative

One of the most blatant examples of racism encountered within the healthcare system was the stereotype linking Indigenous identity to alcoholism. It is a fairly common stereotype among Canadians that “all native people drink”. Participants shared stories of having to deal with this stereotype in their interactions with healthcare providers. The following quotes are examples:

They see your face and they assume that you are an alcoholic. My dad went up [nearby town hospital] and he had this woman doctor. And she’s like, ‘I don’t know why you come up here. This is not your heart, it’s because you drink too much.’ My dad hasn’t had a drink in, I don’t know how many years.(D012)

First time I got a kidney stone attack and I went into the hospital… first thing the nurse asked me was ‘what did you drink?’ Just assuming I drank alcohol right away, that’s a stereotype and racist remark, it’s awful.(E019)

With the next quotation, the participant illustrates how the ‘drunken’ stereotype can directly interfere with provision of quality healthcare:

My husband, we went to the hospital in [city] and he was throwing up blood. He was sick. They kept saying, ‘when was the last time he drank?’ I’m like, ‘he didn’t drink yesterday or in months. It has nothing to do with that. Something is wrong with his stomach. He’s throwing up and there’s blood in it’.(FFG1-3)

### 4.6. Poor Quality Service Attributed to Racism

In discussions on racist encounters within the healthcare system, participants shared many stories regarding poor quality of care. The stories seem to suggest that poor quality care was a manifestation of racism, i.e., that the providers didn’t feel the patients deserved the level of good care that a non-Indigenous person would expect. The first example is by a participant sharing a story about a woman who came to the nursing station for medical support but was abruptly turned away undiagnosed and with dire consequence:

The nurses at the local nursing station] don’t take stuff seriously and they even fired a nurse because someone passed away. When she was trying to come to see [nurse] and she told her to come back in the morning that woman didn’t make it to the morning. Like, if she only took the time to see her she probably still would have been alive.(CFG1001)

The participant in the next excerpt uses strong language to show a violent disregard to her mother’s health by the hospital:

My mom, we just about lost her five years ago. She had an abscess and she’s a diabetic. Everything should have been in her file. They cut her open and they butchered her and it got infected. Sending her back and forth, she ended up in hospital for six months. We almost lost her and since then she hasn’t been able to walk.(D012)

In the next quotation, poor quality care is revealed through the participant’s perception that the healthcare provider lacked interest in communicating with her as an equal on the topic of the health of her children:

The healthcare services, I would say, being a single mother coming here to them, coming or phoning here to the health centre [on-reserve], the nurses make you feel like you’re little, they don’t give you a chance to speak or they make you feel like you don’t know that your child is sick and like, my kid is sick and they don’t give me a chance to explain what is wrong… So I feel that the nurses have to be more caring for the people on the other side. And, that’s my experience for a long time already.(C014)

It is difficult to tell what the underlying intentions were, in the following quotation, for the nurse to have sent the patient home without proper medication for her infection. However, it is important to note that the patient felt she was left uninformed about her health. Did the provider believe the patient would not comply with a prescription given to her? Did she not have time to adequately treat or communicate treatment to the patient? There is not enough information given in the story to know what thinking or belief systems guided the nurse’s behaviour. However, it is evident that the participant speaking felt disregarded and she told this story in response to a question about experiences of racism in the healthcare system.

This one nurse, this was like a month ago, I went and she said I might have a bladder infection I didn’t know what she meant by that and she didn’t explain. She didn’t give me no Tylenol, no nothing, and I just went home and I was like, I didn’t think anything of it. Then a nurse called me 1 ½ week later and she said, “I’m reading through your charts you have a bladder infection are you taking anything for it?” I said no. I said that the nurse just said that I might have it and then she said that they don’t have time to give me anything, like she was in a rush at that time I don’t know, [first nurse] should have taken her time to give me the treatment for my bladder infection instead of like trying to be in a rush.(CFG101)

Some of the participants said that in their interactions with healthcare providers, they felt they had to defend themselves against different types of accusations, e.g., that they indeed were not drinking alcohol prior to coming to the hospital, as suggested in the previous quotes. But also, petty accusations were directed at the participants, such as the one in the following story:

I was having a bad [allergic] reaction and then I phoned and I was accused, the nurse, she didn’t even let me to say who I, I didn’t give my full name or my first name and she was, “oh, you’re phoning here again, just how much Benadryl do you need? You’ve phoned already.” And I said, “Excuse me, I didn’t phone.” And she said, “Oh no, I know you, I recognize your voice.” And I said, “I’m sorry to bother you. I was not seeking for medication”(GFG004-2)

The next story reveals issues that pertain to a lack of understanding of the pace, nature and cultural manners of the people in whose communities the providers are practicing:

We had some incidents with the foot care nurse that she kind of got rough with patients Well, she comes in like every 3 or 5 months She kind of did like the toes or whatever, like the snip snap or whatever or accidentally, she caused injuries She was rushed all the time and people like their feet soaked first, she used to just start, dry, you know, not soften the nail or anything. She just did it and she was nipping, drilling skin just, she was coming in really in a fast pace and she was just trying to get her job just in and out and then people were scared to approach like the service they were getting because it was a professional coming up from the south and which she was hired by our community and expected to do a good job but a lot of the clients don’t even speak because they don’t want, especially our Elders, they don’t want to seem like they’re being ungrateful meanwhile they got an infection happening in their toe, they don’t want to tell on that person and nothing was charted… so it makes it look like the client did it to themselves.(CFG2 4)

The participant below expressed a disconnection with healthcare practitioners that stemmed from aloofness or a coldness of attitude/style:

I went to doctors been to a dentist eye appointments. I feel it’s a matter of how these professions are. I find them to be cold. They are polite, but I’ve often felt they are a cold profession, professional attitudes towards me. I would have to say, the majority off-reserve doctors and dentists, they are like that, at times it feels like the doctor is not really interested in what I’m saying. He didn’t really make eye contact, he more or less just had his paper, his prescription pad and he was writing away but I felt that he really wasn’t listening to me.(E002)

The next two excerpts reveal a direct connection between quality of care and participant feelings that the providers are racist towards them:

Quality healthcare is being treated fairly and like you really matter to somebody, and not being shrugged off as another Native person wanting drugs.(E019)

When I come in for health treatment, I would just like to be treated as a human being.(FFG1-2)

### 4.7. Seeking Racial and Cultural Similarity

In response to blatant racism and poor quality of service, participants said they would try and avoid seeking out healthcare at nursing stations and hospitals. For example:

I don’t go often to the nursing station, just once in a blue moon. I’d avoid it whenever I can, just if it’s really serious or something like that.(A029)

I don’t really go to the hospital down there unless I like really have to go because the doctor down there, she makes me feel like I’m wasting her time.(D004)

Participants also noted racial differences among healthcare providers. In the following quotations, participants revealed feelings of safety and improved service on the part of First Nation, as opposed to non-Indigenous healthcare providers:

I went to see [First Nation Home Care nurse] and she looked at me right away. Sometimes, I think, I don’t like to discriminate, but your own people know what you want and what you need. Like some nurses that are Native, they can pretty well understand, some, not all of them.(C010)

Our people, our health staff are understanding, a little more caring about our own A lot of time, we know everybody and who the family members are, where in our First Nations, we are linked. So we have big families, connections and so the level of actual care the feelings for wanting to help somebody are often [stronger].(E004)

First Nation healthcare providers, they give care and seem to care more. Like it doesn’t seem racist or they brush the Natives off that easily cause, we respect our people, our Elders Off reserve, they’re kind of just rush through send you away.(H018)

The service we receive from our Indigenous staff would be best.(D014)

Few opportunities for meaningful engagement and familiarity between First Nation and non-indigenous Canadians (healthcare providers).

A resounding theme discovered in the data, was the lack of connection, or more specifically, the limited opportunities for meaningful engagement and familiarity between First Nation and non-Indigenous peoples. This theme was revealed in several different scenarios. For example, with regards to traveling to the city for medical appointments, participants noted:

It’s a culture shock for us.(GFG004-3)

Different people, different manners(GFG004-11)

Participants said healthcare providers who have not taken the time to get to know the community history, values, and customs were showing disrespect and a disregard that might be conceived of as a feeling of superiority over their patients, and the community in general. A lack of knowing one another perpetuates separation between ‘races’:

Over here [in the community] nurses pretty much understand and community [out] there, you try and talk to some of the healthcare providers and they don’t really understand some of the cultural components of it, they don’t understand the community they don’t know what all we’re going through.(E015)

[Nurses who come to the community for short-term contracts, a practice prevalent in the federal workforce] they don’t know your history, right? They don’t really know your history; they just come from the city.(GFG004-6)

They don’t know the clients. They don’t know the community and then the clients or the patients have to tell their whole story again and again, so the consistency is not there when we have those nurses coming in.(GFG004-1)

Alternately, participants noted feeling as outsiders when traveling out of the communities for medical appointments in the cities.

You get looked at for being First Nation because we get our medication paid for and sometimes you hear comments about stuff like that off the reserve. So at home, you see more caring.(E004)

### 4.8. A Lack of Consideration of Accessibility to Healthcare Issues for First Nations

Systemic exclusion from the technologies and expertise of the healthcare system were evident in the stories. Particular emphasis was placed on the lack of consideration on behalf of the system of social and economic obstacles to accessibility that the First Nations face, especially with regards to having to travel far distances in order to access healthcare. The following excerpts are illustrative of this point:

They don’t consider how long it takes and how far we have to travel to get to our appointments.(D013)

It’s hard to go Winnipeg because for a lot of those services they don’t accommodate you [when] you’re from an isolated community.(C011)

Healthcare provided me with travel to my appointment in Winnipeg but my stay there wasn’t covered.(D001)

(We are) sent out of the reserve to go to your appointment of town you usually get sent out there and they don’t put us in a place to relax when you’re ill, totally ill.(G002)

I was sent (to Winnipeg); by myself I go for whatever appointment I have then they’re going to shove you out from there, to another section, drag you around… What are you supposed to do? We’re in a strange place I don’t know my way around the city.(G002)

Like if I’m going to Winnipeg [for a medical appointment] I’ll have to worry to get into housing or food and I’ll end up going to food banks and that’s bad.(D009)

It’s scary out there when they do that to us, you know Some Elderly people are lost in the city. They don’t know the language and they don’t know where they’re going.(G002)

## 5. Discussion

Evidence collected for the study is consistent with the literature on racial discrimination against First Nations in the healthcare system. Data was organized into 9 interconnected themes, which together lend a greater understanding of the root causes and factors that perpetuate racism. As well, the themes reveal the multiplicity of ways racism acts to negatively affect the health and wellness of First Nations peoples and to contradict their cultures, languages, institutions and overall governance. Further, the thematic structure presented promotes discussion on the need to eradicate racism and to create an equitable and just healthcare system that prioritizes First Nation interests.

Participants talked about issues regarding distrust between them and non-indigenous individuals and institutions, i.e., the Regional Health Authority, stemming from previous interactions with community members. Within these stories, it was also evident that the participants felt the services that were offered were not fitting or contradicted First Nation values, interests and priorities.

Navigating a system that devalues First Nation health and wellness, speaks to political and economic concerns that take precedence over First Nation wellbeing. Participants shared stories where financial and jurisdictional complications stood in the way of improved healthcare. These issues have been studied previously [60,61]. In the current study, participants revealed the personal impacts of unresolved or blurred policy and imbalanced political and economic interests. A realignment of the goals of the Canadian healthcare system is necessary in order to ensure that the system is working to enhance the health of the Canadian population equitably and in a diverse enough way as to consider cultural, geographic, socio-economic and historic-political realities of the First Nations.

The lack of consideration for First Nation realities is revealed in the healthcare system’s preference for English and biomedicine over accommodation of First Nation languages and traditional medicines. Through their stories, participants showed how the onus was upon them to adjust to biomedicine, rather than the reverse. Participants offered suggestions, i.e., incorporation of advocates and translators, accommodating escorts for medical appointments, and including traditional medicine in healthcare.

Colonization and systemic racism continue to affect the everyday lives of people [62]. Through the articulated struggles, we can hear the very real and ongoing consequences of a lack of resolution of Canadian and First Nation relations. In his paper on settler colonialism and elimination of the native, Wolfe [63] writes that colonialism first destroys in order to replace. Invasion is a structure, not an event. With identities redefined and made to fit into the structural imperatives of a new nation, it comes as no surprise that the participants would express a lack of trust or comfort that their health needs were being adequately met. When participants shared that the healthcare system failed to meet their needs and that the system is not as it is said ‘accessible to all’ they were highlighting not only the fact of its racist and colonizing structure, but also that Canada has not progressed far enough since the Royal Commission on Aboriginal Health highlighted inequities in healthcare more than 25 years ago [64].

Healthcare as a right of citizenship is also highlighted. A right to healthcare, covered by the Canada Health Act (CHA) implies the right to high-quality healthcare for First Nation peoples on par with the rest of society. The CHA established criteria and conditions related to insured health and extended healthcare services that the provinces and territories must fulfill in order to receive the full federal cash contribution under the Canada Health Transfer. The CHA is to ensure that all eligible Canadian residents have reasonable access to insured health services on a prepaid basis, without direct charges at the point of service. As Canadian citizens, First Nations have the right to equitable healthcare without prejudice or discrimination. The current imbalance within healthcare that leaves First Nations dependent upon the political and economic whims of a more powerfully resourced group within Canadian society is contrary to the foundation of the CHA, an imbalance which is felt in the one-on-one interactions with the healthcare system, described by the participants.

Among Canadian healthcare providers, there are those who are continuing to stereotype First Nation peoples by insinuating that since patients are Indigenous, they must also drink alcohol. The history of alcohol use among First Nation peoples in Canada is complex. Chelsea Vowel, a Cree lawyer and Indigenous rights activist from the University of Alberta writes [65]:

Alcohol was a mainstay of trade goods after Contact, and then for a time the sale and provision of alcohol was banned completely for the ‘protection’ of Indigenous peoples, resulting in a booming and unhealthy underground trade. The idea that Indigenous peoples are helpless to resist the lure of alcohol, that we are genetically weak and more susceptible to it, plays into the notion of our supposed inferiority.

Further, Vowel writes, “to really get a sense of the problem, you have to understand the history of colonization in this country.” Colonization is responsible for some very real problems with addictions among First Nation peoples as many communities continue to struggle with high rates of alcoholism, drug addictions, as well as fetal alcohol addiction in children. Root causes are well documented, including residential schools, the Indian Act, health and social welfare policies and practices, geographic displacement and isolation, racism, and intergenerational trauma [66].

However, the assumption that a First Nation person is coming into the hospital with an alcohol problem is flawed logic, ‘not based in fact or rational thinking’ [67] and to an extent that the stereotype interferes with a proper diagnosis and, therefore, treatment, can be dangerous, as revealed through the stories. The connection between quality of care, miscommunication (i.e., unintentionally making race-based blaming assumptions) and optimal treatment has also previously been established [68].

The stereotype persists notwithstanding statistical evidence to the contrary. For instance, although binge drinking is higher among those First Nation peoples who are drinking, more First Nation peoples abstain from drinking, and those who do drink, do so less frequently than the rest of Canadians [69] (pp. 76, 81).

The research participants provided many examples of inadequacy in healthcare, from rushed to neglectful treatment and a lack of proper diagnosis. Further, issues of accessibility and a lack of communication on the part of the medical staff were also recorded. Quality intersects with racism in the participant beliefs that they would have been treated better if they were not indigenous. Racial discrimination and racial profiling have previously been correlated with poor physical and mental health [70]. Further, these findings suggest that direct and persistent vicarious racial discrimination are detrimental to health [71]. The connection made in the current study between quality of healthcare provided and racism is important as it raises issues of unresolved stereotypes, persistent racism, and poor overall relationships between First Nations and non-indigenous Canadians.

Individual responses to racism and perceived racism varied. Participants said they would avoid going to nursing stations and hospitals ‘where possible’. Avoidance can be a dangerous response, particularly in instances that call for immediate intervention. Seeking safety in racial and cultural similarity was another response to perceived racism and poor quality of care by non-indigenous healthcare providers. Indigenizing healthcare by increasing numbers of Indigenous healthcare providers makes sense and it has been a focus of the Canadian governments and universities in order to improve healthcare and, ultimately, the health of First Nations [72]. Increasing support and workforce numbers of Indigenous physicians does not excuse an emphasis on improving understanding and relationships between First Nation and non-indigenous healthcare providers.

Participants shared examples of a lack of consideration on the part of Canadian society for the real circumstances of First Nation peoples, e.g., socioeconomic and geographic obstacles to healthcare. Participants noted the psychological impacts of public misconceptions, i.e., that First Nation peoples get healthcare ‘freebies’. Discussion among Canadians and correct information through our education system can curb this negativity and increase feelings of comfort for individuals and families travelling from First Nation communities to cities for medical appointments. The will for change must come from the Canadian government, as Richmond and Cook have suggested [73], by way of an informative and healthy public policy that recognizes and prioritizes the rights of Canadian First Nation peoples to achieve health equity.

Finally, biomedicine and the Canadian healthcare system deliver piecemeal services to First Nation communities, which actually run contrary to a worldview that sees whole people, interconnected within families and environments. Assumptions that healthcare can be responsible only for delivering a technological service and treating individual illnesses, can leave First Nation peoples in dire situations, particularly when travelling outside of their communities for healthcare.

The racist structures, upon which medicine and the healthcare system are built, are developed in official discourses, which when left unchecked provide the substance for which the status quo is maintained. This finding is consistent with Mathews’ [47] remark that Canadian healthcare is founded on systemic racism through violent unilateral imposition of Canadian social, economic, cultural and political dominance over Indigenous land and lives under section 92(24) of the Constitution Act, 1987, Indians, and Land Reserved for the Indians, as well as under Indian Act 1985. When Canadians, the healthcare system, and biomedicine assume only piecemeal responsibility for services to First Nations, or that First Nations are responsible for their own socioeconomic and geographic barriers to medical provision—it is because we are lacking substantial critical analyses of our institutional protocols and discourses, which causes them to appear self-evident, true, empirically valid and morally desirable.

Evidence for systemic racism is seen in the normalization and legitimization of an array of dynamics—historical, cultural, institutional and interpersonal—that routinely advantage non-indigenous Canadians while producing cumulative and chronic disadvantages to First Nation peoples [40]. A lack of consideration to the circumstances of First Nation peoples is, in other words, a system of hierarchy and inequity, where those who are not from the First Nations receive preferential treatment.

Bringing together participant stories on rights and traditions, in particular, the need to reclaim and revive the traditional medicines, we can see that the time has come not only to dismantle but also to modernize the colonial superstructure. As Maureen Lux so poignantly revealed in her book, *Separate Beds*, the history of segregated ‘Indian’ hospitals in healthcare policy since the 1920s has revealed the limits of Canadian liberal democracy.

Colonized in the interests of the nineteenth-century state formation, Aboriginal people were marginalized in the construction of the welfare state. Lines of segregation and isolation, tended by the power of the state and rationalized in the language of medical humanitarianism, did not negate Canada’s liberal democratic values. In fact, they were integral to the formation of national health and normal white citizenship.[43] (p. 197)

We agree with her argument that Canada and its colonizing and white supremacist policies must be held accountable for the damages they have and continue to cause at all levels and to do away with healthcare practices that fail to put the health interests of the First Nations *first*.

Some limitations to the study exist. The study was conducted in partnership with a sample of Manitoba First Nation communities. Findings are not readily generalizable to other First Nation communities in Manitoba or in other Canadian provinces. Further research is necessary to understand experiences of racism in other First Nation communities. It can also be noted that although principles of GT and decolonizing research emphasize induction, inclusion, reflexivity, and, in terms of the latter, a relinquishing of scientific control, the study is nevertheless academic at its roots. Community members within the communities who did not participate in the study may or may not share the same ideas and experiences of those who were participants. It is important, however, that the community residents’ comments in this study do reflect several of the major issues which have gained prominence regarding systemic racism in healthcare across Canada in the autumn of 2020.

## 6. Conclusions

The time is long overdue for us to embrace a more inclusive, diverse and equitable social structure and to reshape our health institution so that it reflects such ideals. A healthy healthcare system is one that builds upon the strengths of First Nation communities and returns governance under their control. Within such a perspective, we will see possibilities for health unavailable under colonialism.

This paper is written in a time where, nationally and globally, societies are speaking out against racism, asking for change. The participants of this research have given concrete examples where change is warranted. The findings of this study can be expanded upon and used to redevelop the healthcare system into one that is integrative and considers Indigenous people, their right to culture, to freedom, and to health.

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
