# Peer review of "Confronting Racism within the Canadian Healthcare System: Systemic Exclusion of First Nations from Quality and Consistent Care"

_ijerph, 2020, doi:10.3390/ijerph17228343_

Round 1

Reviewer 1 Report

See comments on the document.

Author Response

Thank you. Please see the attachment, with your suggested revisions included.

Reviewer 2 Report

This is a sound paper whose foundation on grounded theory is supported with a meticulous social sciences analytic approach. The interviews support the conclusions reached. It is well-written and presented in Standard English without mixing it with American English.

There are a few minor editorial corrections to take care of:

p. 2, line 56: place a comma after "social justice" to separate it from "healthy diversity."

p. 2, line 65: place a period at the end of "et al," to read "et al.,"    Also see p. 5, lines 142. 154; p.6, lines 180, 186, 189, 190; p. 7, line 213; p. 8, lines 250, 265, 268; p. 21, line 737.

p. 3, line 89: "with more subtly" should read "with more subtlety..." Line 92: "a peoples" should read "a people"; Line 93: place a comma after "her book"; delete "on", and italicize "White Fragility" to read "her book, White Fragility,". Line 95: change "it's" to "it is" to make it more formal.

p. 4, line 130: change "as least" to "at least..."

p.8, line 272: remove the comma after "following" and place the comma after "times."

p.9, line 304: change "lead..." to "led...

p.10, line 348: change "them..." to "then..."

p.11, line 357: make the idea more meaningful by inserting the following phrase  in brackets before "sent out..." to read "[we are] sent out..." Line 366: "moths..." should read "months..."

p. 12, line 401: "most proudest..." should read "proudest..." Line 420: change "quality of healthcare" to read "quality healthcare."

p. 16, line 531: italicize the whole sentence. Line 541: change "haven't" to "have not" to make it more formal.

p.17, line 566: change "excepts..." to "excerpts..." 

p. 18, lines 616-617 should be corrected to read as follows: In his paper, "Settler Colonialism and the Elimination of the Native," Wolfe... Line 623: delete "too" after "progressed" and insert "enough" after "far..." to read "progressed far enough..." 

p.19, line 635: change "on-on-one.."to read "one-on-one..."

p. 21, line 725: delete colon after "book" and replace with comma to read, "book,..."

Your references are very good but you should be more painstaking in editing the entries. The main problem is that you neglected placing periods after the authors' initials, e.g. line 764: Browne AJ instead of Browne A.J. and his co-authors. Also see lines 768,779, 797, 808, 810, 816,820, 831, 838,841, 853,869, 881, 892,900, 909, 911, 931, 932.

On line 751, "f" should be upper case letter in "Foundation."

line 761: shrink the space in "redressing  racism"

lines 808-809: remove underline

line 811 should be corrected and written/italicized appropriately

line 813: shrink the space between Inuit   Health

lines 827, 835, 923: remove "et al." and give the names and initials of all the authors. This is where you give the authors their due credit instead of keeping them in the anonymity of et al.

line 832: change "Ethn" to "Ethnic

line 857: delete all those three dots before "Safarov"

line 859: delete the extra period after 'sheet'.

line 917: shrink the space in "Peoples   (1996)."

This is a good paper overall. Go through every line of the References and edit them appropriately.

Author Response

Thank you. Please see the attachment with your suggested revisions included.

Reviewer 3 Report

Dear authors,

Please see the attached document for my comments and suggestions.

Thank you for your important work.

Author Response

(The authors gave the same response as above.)
